# A Magnetoelectric Automotive Crankshaft Position Sensor

**DOI:** 10.3390/s20195494

**Published:** 2020-09-25

**Authors:** Roman Petrov, Viktor Leontiev, Oleg Sokolov, Mirza Bichurin, Slavcho Bozhkov, Ivan Milenov, Penko Bozhkov

**Affiliations:** 1Institute of Electronic and Information Systems, Novgorod State University, 173003 Veliky Novgorod, Russia; s181607@std.novsu.ru (V.L.); Oleg.Sokolov@novsu.ru (O.S.); mirza.bichurin@novsu.ru (M.B.); 2Faculty of Machinery and Construction Technologies in Transport, Todor Kableshkov University of Transport, 1113 Sofia, Bulgaria; stbozhkov@vtu.bg; 3Faculty of Telecommunication and Electrical Equipment in Transport, Todor Kableshkov University of Transport, 1113 Sofia, Bulgaria; milenov55@abv.bg; 4Olimex Ltd., 4002 Plovdiv, Bulgaria; info@olimex.com

**Keywords:** magnetoelectric effect, magnetoelectric structure, magnetoelectric sensor, crankshaft position sensor, automotive sensor

## Abstract

The paper is devoted to the possibility of using magnetoelectric materials for the production of a crankshaft position sensor for automobiles. The composite structure, consisting of a PZT or LiNbO_3_ piezoelectric with a size of 20 mm × 5 mm × 0.5 mm, and plates of the magnetostrictive material Metglas of the appropriate size were used as a sensitive element. The layered structure was made from a bidomain lithium niobate monocrystal with a Y + 128° cut and amorphous metal of Metglas. Various combinations of composite structures are also investigated; for example, asymmetric structures using a layer of copper and aluminum. The output characteristics of these structures are compared in the resonant and non-resonant modes. It is shown that the value of the magnetoelectric resonant voltage coefficient was 784 V/(cm·Oe), and the low-frequency non-resonant magnetoelectric coefficient for the magnetoelectric element was about 3 V/(cm·Oe). The principle of operation of the position sensor and the possibility of integration into automotive systems, using the CAN bus, are examined in detail. To obtain reliable experimental results, a special stand was assembled on the basis of the SKAD-1 installation. The studies showed good results and a high prospect for the use of magnetoelectric sensors as position sensors and, in particular, of a vehicle’s crankshaft position sensor.

## 1. Introduction

Improvement of automobiles is an actual problem for a large number of scientists and engineers, which they are engaged in to improve the consumer properties of machines and increase their reliability and safety [1]. We are exploring all the new components that provide better performance and lower costs. One of the important components of a modern car is sensors. Sensors must provide accurate and timely information for the on-board computer system and actuators. Sensors are used in almost all vehicle systems. In the engine, they measure the temperature and pressure of the air, fuel, oil, and coolant. Many moving parts of the car (crankshafts, camshafts, throttles, shafts in the gearbox, wheels, and exhaust gas recirculation valves) are connected to position and speed sensors. A large number of sensors are used in active safety systems [2,3,4,5]. One of these sensors is a position sensor. In particular, it is used in the ignition system as a crankshaft position (CKP) sensor. In modern cars, such sensors operate on the induction effect, or the Hall Effect [6,7], and is possible on giant magnetoresistance [8,9]. Sensors operating on the magnetoelectric effect (ME) also have been proposed [10,11]. A significant improvement of such sensors will be a new ME structure that does not require magnetization for its operation. Based on such a structure, it is possible to construct sensors that do not contain permanent magnets or solenoids for magnetization.

Today, innovations in electronic technology are preceded by new materials. The use of functional materials now known allows us to develop new components and devices whose physical and chemical properties are sensitive to changes in the environment, for example, magnetic field and temperature, and so on [12,13]. Consider one of the types of functional materials—magnetoelectrics. The main physical property of magnetoelectrics is the ME effect [14]. The ME effect is observed both in single-phase structures, in two-phase structures, and in various others. In general, this type of functional material is obtained using magnetostrictive and piezoelectric structures. A number of devices based on magnetoelectrics are presented in Table 1.

As can be seen from the table above, today ME materials are widely used in various applications [15,16,17,18]. The main focus of this article is on a position sensor, the sensitive element of which is the ME composite layered structure.

Structurally, the CKP ME sensor is an ME element with an electronic signal processing circuit located in the immediate vicinity of the crankshaft pulley or a disk with magnets fixed to it. The rotation of the shaft leads to the creation of an alternating magnetic field in the immediate vicinity of the sensor. The sensor processes the received variable signal and transfers it to the on-board computer of the car for further analysis. The use of ME position sensors is also possible for the analysis of other processes where there is movement of mechanical parts, or there is a constant or alternating magnetic field. Improving the characteristics of the CKP ME sensor is an actual and popular task for research.

Magnetoelectrics are materials that have both magnetic and electrical ordering. The interconnection of the magnetic, electrical, and elastic properties of multiferroics leads to the fact that cross effects are possible in them, linking the magnetic and electrical characteristics of the material. When an external electric field is applied to such a structure, a change in magnetization occurs, and vice versa, when an external magnetic field is applied, a change in polarization occurs. This effect is called magnetoelectric. ME sensors have a number of advantages over traditional ones; in particular, they have a higher sensitivity, smaller dimensions, and a lower mass, as well as ensuring operation without additional energy sources.

Previously, experimental and theoretical studies of the use of a composite structure based on a magnetostrictive-piezoelectric as a sensitive element for a CKP sensor, based on microfiber composites and on the basis of a self-magnetizing gradient structure using Nickel, were carried out; the results of all these works were positive. However, for the application of a CKP ME sensor in cars, it is necessary to further improve them, increasing the sensitivity and other characteristics.

The purpose of this work is to study the ME automotive crankshaft position sensor based on bimorph lithium niobate. The use of a composite structure based on lithium niobate will significantly increase the magnitude of the ME effect [19], increase the sensitivity of the device, and reduce the noise level, while there also is no need for magnetization by an external constant magnetic field, which greatly simplifies the possibility of application in automotive systems.

## 2. ME Structure

The ME structure is an integral part of the position sensor. In order to determine the optimal mode of the sensor operation and the acceptable ME coefficient value, a comparison of different ME structures was performed. A lithium niobite monocrystal (the author team expresses their sincere gratitude for the sample of bidomain lithium niobate monocrystal with a Y + 128° cut kindly provided by Andrei V. Turutin, National University of Science and Technology MISIS, RF) and PZT ceramics were used as the piezophase, and Metglas was used as the magnetostrictive phase of the ME structure. Studies of symmetrical and asymmetrical structures of the ME was carried out. In this article, an asymmetric structure refers to a structure that has a different thickness of the magnetostrictive layer in the upper and lower plates. In particular, an asymmetric structure is a structure that has a magnetostrictive layer on only one side of the element.

Figure 1 shows the position of the interface between the lithium niobate and Metglas relative to the neutral plane in the ME composite with copper or aluminum as the lower layer. If Metglas is used as the lower layer and the ME structure of the composite becomes symmetrical, then ${}^{m}t$ must be used instead of the thickness ${}^{s}t$.

${}^{p}t={}^{p1}t+{}^{p2}t$ is the thickness of the bimorph piezoelectric; ${}^{p1}t$, ${}^{p2}t$ are the same thickness opposite each other of the polarized bimorph layers of lithium niobate; $t={}^{p}t+{}^{m}t+{}^{s}t$ is the thickness of the ME composite layered structure; ${}^{m}t$, ${}^{s}t$ are the thicknesses of the Metglas and bottom layer, respectively; and $\rho =\left({}^{p2}\nu +{}^{p2}\nu \right){}^{p}\rho +{}^{m}\nu {}^{m}\rho +{}^{s}\nu {}^{s}\rho$ is the effective composite density; ${}^{p1}\nu ={}^{p2}\nu ={}^{p1}t/t$, ${}^{m}\nu ={}^{m}t/t$*,*
${}^{s}\nu ={}^{s}t/t$ are the volume fractions of the piezoelectrics, ferromagnetic, and lower layer.

For asymmetric structures with a copper or aluminum electrode:(1)$$\langle q_{11}\rangle =\frac{{\bar{q}}_{11}\left(2z_{0}+{}^{m}t\right)}{2{\;}^{m}t},$$
where ${\bar{q}}_{11}={}^{m}Y{}^{B}q_{11}$ the position of the interface of the piezoelectric and Metglas relative to the midline of the composite:(2)$$z_{0}=\frac{{\bar{c}}_{11}^{D}{\;}^{p}t{}^{2}+2{\;}^{s}Y{\;}^{s}t{\;}^{p}t+{}^{s}Y{\;}^{s}t{}^{2}-{}^{m}Y{}^{B}{\;}^{m}t{}^{2}}{2\left({\bar{c}}_{11}^{D}{\;}^{p}t+{}^{m}Y{}^{B}{\;}^{m}t+{}^{s}Y{\;}^{s}t\right)},$$
where
(3)$$\begin{array}{l} {\bar{c}}_{11}^{D}={\left({}^{p}s{}_{11}^{E}-\frac{{}^{p1}d{}_{31}^{2}}{\varepsilon_{33}^{T}\varepsilon_{0}}\right)}^{-1}\\ {}^{p1}\bar{h}{}_{31}=\frac{{\bar{c}}_{11}^{D}{\;}^{p1}d{}_{31}}{\varepsilon_{33}^{T}\varepsilon_{0}}\\ {}^{p2}\bar{h}{}_{31}=\frac{{\bar{c}}_{11}^{D}{\;}^{p2}d{}_{31}}{\varepsilon_{33}^{T}\varepsilon_{0}}\\ {\bar{\beta }}_{33}^{S}=\frac{1+{\;}^{p1}\bar{h}{}_{31}{\;}^{p1}d{}_{31}}{\varepsilon_{33}^{T}\varepsilon_{0}} \end{array},$$
(4)$${}^{m}Y{}^{B}=\frac{{}^{m}Y}{1-{}^{m}K{}_{11}^{2}},$$

${}^{p}s{}_{11}^{E}$—piezoelectric compliance at a constant electric field strength; ${}^{p2}d{}_{31}=-{}^{p1}d{}_{31}$—piezoelectric coefficients of the two layers of the bimorph niobate lithium; $\varepsilon_{33}^{T}$—relative dielectric constant of lithium niobate at a constant mechanical stress; ${}^{m}K{}_{11}^{2}={}^{m}Yq_{11}^{2}/\mu \mu_{0}$—squared coefficient of the magnetomechanical coupling; and ${}^{m}Y$, $q_{11}$, μ—Young’s modulus, the pseudo-piezomagnetic coefficient, and relative magnetic permeability of the Metglas.

For symmetrical structures:(5)$$\langle q_{11}\rangle =\frac{\left({\bar{q}}_{11}^{u}-{\bar{q}}_{11}^{d}\right)\left({}^{p}t+{}^{m}t\right)}{2{\;}^{m}t},$$
where ${\bar{q}}_{11}^{u}$ is the pseudo-piezomagnetic coefficient of the top Metglas layer and ${\bar{q}}_{11}^{d}$ the pseudo-piezomagnetic coefficient of the lower Metglas layer
(6)$$z_{0}=\frac{{}^{p}t}{2},$$

Take into account that
(7)$$\langle \beta_{33}^{S}\rangle =\frac{1}{{}^{p}t}\int_{z_{0}-{}^{p}t}^{z_{0}}{\bar{\beta }}_{33}^{S}dz={\bar{\beta }}_{33}^{S},$$
(8)$$\langle c_{11}\rangle =\frac{1}{t^{3}}\left(D-\frac{{}^{p}t{}^{3}{\langle h_{31}\rangle }^{2}}{\langle \beta_{33}^{S}\rangle }\right),$$
(9)$$k={\left(\frac{\rho }{t^{2}\langle c_{11}\rangle }\omega^{2}\right)}^{\frac{1}{4}},$$
where $D={}^{p}D+{}^{m}D+{}^{s}D$—the full cylindrical stiffness of the composite beam,
(10)$$\begin{array}{l} {}^{m}D=\frac{1}{3}{}^{m}Y{}^{B}{\;}^{m}t\left({}^{m}t{}^{2}+3{\;}^{m}tz_{0}+3z_{0}^{2}\right)\\ {}^{p}D=\frac{1}{3}{\bar{c}}_{11}^{D}{\;}^{p}t\left({}^{p}t{}^{2}-3{\;}^{p}tz_{0}+3z_{0}^{2}\right)\\ {}^{s}D=\frac{1}{3}{}^{s}Y{\;}^{s}t\left({}^{s}t{}^{2}+3{\;}^{s}t\left[{}^{p}t-z_{0}\right]+3{\left[{}^{p}t-z_{0}\right]}^{2}\right) \end{array},$$

For this method of fixing, the calculation gives
(11)$$\alpha_{ME}=\frac{{}^{m}t{}^{2}{\;}^{p}t{}^{2}\langle q_{11}\rangle \langle h_{31}\rangle \langle \beta_{33}^{S}\rangle \left(3r_{1}r_{3}-2r_{1}-2r_{3}+1\right)}{t\left[\langle c_{11}\rangle klt^{3}\langle \beta_{33}^{S}\rangle \left(r_{1}r_{4}-r_{2}r_{3}\right)-{}^{p}t{}^{3}{\langle h_{31}\rangle }^{2}\left(3r_{1}r_{3}-2r_{1}-2r_{3}+1\right)\right]},$$
where $\langle h_{31}\rangle ={}^{p1}\bar{h}{}_{31}/4$, $r_{1}=\cosh\left(kl_{4}\right),\;r_{2}=\sinh\left(kl_{4}\right),\;r_{3}=\cos\left(kl_{4}\right),\;r_{4}=\sin\left(kl_{4}\right)$.

The permanent magnet is centered relative to the midline of the symmetric composite sample with some error; therefore, the magnitudes of the magnetizing constant field in the three upper and three lower layers of the Metglas are slightly different. Because of this, the pseudo piezomagnetic coefficients in the upper and lower layers of the Metglas are slightly different. Therefore, the effective pseudo-piezomagnetic coefficient $\langle q_{11}\rangle$ of the composite is slightly different from zero. This explains the presence on the graph of a small peak in the bending EMR region for a symmetric composite sample.

## 3. Experimental Results

The multiferroic layered structures were made from a bidomain lithium niobate monocrystal with a Y  +  128° cut and amorphous metal of Metglas. The material showed high values of the ME coefficient on the bending mode of oscillations (shown in Figure 2), which is necessary for a good-quality position sensor.

The size of the ME element structure is 20 mm × 5 mm × 0.5 mm. Three different structures were investigated to improve the working of the ME element on the bending mode. The first structure is symmetrical (three layers of Metglas/LiNbO_3_/three layers of Metglas). The thickness of one layer of Metglas is about 23 microns. The second structure is asymmetric (three layers of Metglas/LiNbO_3_/one layer of copper). The thickness of the copper layer is about 35 microns. The third structure is asymmetric (three layers of Metglas/LiNbO_3_/one layer of aluminum). The thickness of the aluminum layer is about 10 microns. The third sample showed the best results. The value of the ME resonant voltage coefficient was 784 V/(cm·Oe), and theoretical value is 797 V/(cm·Oe). The theoretical and experimental values are in good agreement. This sample was used for further research.

A sample based on piezoceramics was also made. An ME element from a layered composite structure consisting of layers of Metglas (FeBSiC) and piezoelectric PZT was researched. The ME element was 20 mm × 5 mm × 0.5 mm in size. The low-frequency, non-resonant ME coefficient for ME element amounted to about 3 V/(cm·Oe). This structure was symmetrical (three layers of Metglas/PZT/three layers of Metglas).

The results of the experiments have shown that it is possible to use both the ceramic piezoelectric PZT and the monocrystal material LiNbO_3_ for designing the position sensor. At the same time, a significant difference in the level of the ME coefficient was obtained. ME material with a large ME coefficient allows signal processing in an electronic circuit without an amplifier, or it is possible to use the ME structure with significantly smaller dimensions. It is also significant that the use of a monocrystal material removes the technological problems existing in ceramic materials with the repeatability of their parameters. The temperature stability of the sensor is also getting better. The critical Curie temperature for the PZT ceramics is about 290 °C, while the recommended temperature for PZT ceramics of about 200 °C may not be sufficient for technical applications. Since lithium niobate is a ferroelectric with a Curie temperature about 1210 °C, the temperature range of the ME sensor will be limited by the magnetostrictive material and the technological connecting of the structure elements in the ME material. The aging of the ceramic-based piezomaterial is estimated at 10% in two years; the aging of the monocrystal practically does not occur, which means that the degradation of the sensor material based on the monocrystal will be insignificant. For the Metglas alloy the Curie temperature is about 380 °C. The noise of such ME sensors is insignificant, amounting to several pT Hz^−1/2^ according to data published for example in [20]. Thus, in general, the characteristics of the ME structures suggest sustainable function of the ceramic magnetoelectric sensor up to a temperature of 200 °C, and the sensor based on lithium niobate up to a temperature of 380 °C.

## 4. Design of CKP

The properties of ME structure, material composition, manufacturing technology of the structure, and ME element characteristics of the sensor were well described in previous works. The sensor design is shown in Figure 3. The ME element, permanent magnet, and PCB of the electronic circuit are placed in the sensor casing. NdFeB permanent magnets with dimensions of 6 mm × 3 mm × 1 mm were fixed near with ME element on the casing wall. The electronic circuit of the PCB realized the comparator function. The position sensor was installed nearby, at a distance of about 1 mm, from the synchronous disk that was implemented in the form of a steel-toothed ring or a ring with magnets. The Renault 7700103069 CKP sensor was chosen as a prototype that corresponds to the shape and overall dimensions of the developed sensor. This choice was due to the possibility of interchangeability of these sensors without significant changes in the car design. The CKP ME sensor design has been patented.

The structure, properties, and characteristics of the ME material used for the manufacturing of the sensor sensing element are described in detail in previous works. The design and structure of the sensor are shown in Figure 4. The sensor housing is made in the form of a cylinder. An auxiliary beam (2) is located inside the housing. The ME element (4) is fixed to the auxiliary beam. Permanent magnets (5) must be located near the ME element. For the correct operation of the sensor, a scheme for processing the output signal (3) was also developed. The ME sensor must be installed near the synchronous disks, not more than 1 mm. The sensor body is made similar to existing analogues, such as the Renault 7700103069 CKP induction sensor, regarding the overall sensor dimensions.

Compared with an inductive sensor, ME sensors do not have the effects of degradation of the inductor and magnetic core. Compared to Hall effect sensors (where the degradation of the main element is also observed), ME sensors have a significantly lower current consumption. The power supply circuit of the ME sensor is such that the ME structure does not require power, and only several mA is required for the consumption of the signal processing circuit. The sensor prototype has a hybrid microelectronics design containing an ME element, electronic circuit, and magnet, which of course significantly reduces its reliability. It is obvious that the use of single-chip microelectronics technology for ME sensor manufacturing will improve the reliability parameters by several orders, as well as allowing a reduction in the overall dimensions and weight.

The ME position sensor without signal processing can also be applied as a car CKP sensor, which has been proposed in other papers. The difficulty of integrating this sensor into the car’s electronics is the main disadvantage of such a design. Processing of signals received from the ME element with further output to the CAN bus, and then to the car’s ECU (electronic control unit) for solving this problem, is proposed.

Signal forming was used for initial preparation for further processing in the microcontroller. The device block scheme is shown in Figure 4. The meander signal generator starts the process to excite an alternating magnetic field in the solenoid coil. At the same time, an alternating magnetic field in the space around the ME sensor is created during rotation either by the teeth of the steel-toothed ring or by the magnets that are mounted on the ring. Thus, in the location region of the ME sensor, two alternating magnetic fields are excited: one is the low-frequency field with information about the crankshaft speed, and the second is the high-frequency modulating field of the solenoid coil. The maximum ME coefficient is reached when the frequency of the modulating field is adjusted to the frequency of the EMR for the ME element. Further, the ME sensor signal is amplified and processed by the comparator scheme, forming rectangular pulses. Then, the microprocessor processes the received signal and the signal goes through the CAN controller and CAN transceiver to the CAN bus, subsequently going to the automotive ECU.

This solution allows you to ensure the optimal level of the output signal, low noise level, as well as integrate the position sensor in modern automotive systems.

## 5. Principle of Operation

The operation principle of the ME position sensor is based on the ME effect and can be considered on the example of the ME CKP sensor in the test bench. The ME CKP sensor must be located near of the steel-toothed ring (4) of the car’s crankshaft. The steel-toothed ring has rectangular teeth on the outside, which are markers of turning at a certain angle. One marker type determined the periodic combinations of the “tooth/interval between the teeth” and another stretched the interval between the teeth for a marker of the start, as shown in Figure 5. The permanent magnet (2) that is located near the ME element (1) creates a DC magnetic field (bias field). An AC magnetic field induced by the rotation of the ring due to the alternation during the rotation of sections with low magnetic permeability—the “interval between teeth” and high magnetic permeability “tooth”—affects the sensor’s ME element. An ME element for each combination of “the tooth/interval between the teeth” generates a voltage pulse. Conclusions about the crankshaft angular position can be made via the pulse number in the oscillogram of the sensor output characteristics, where the start of the counts will be determined by a special label from the stretched interval between the teeth. The height of the tooth has an impact on the magnetomotive force magnitude due to the magnetic flux passing through the steel-toothed ring. If an ME sensor is located above the tooth then the magnetomotive force F_1_ will be greater (Figure 5a); in the case when the sensor is located between the teeth in the gap (Figure 5b), then the magnetomotive force F_2_ becomes less, since the distance *l*_1_ is less than the distance *l*_2_. Thus, the ME sensor in Figure 5 is exposed to an alternating magnetic field, caused by the rotation of the steel-toothed ring (4) and due to the rotation of the sections with a high magnetic permeability—“tooth”—and those with a low magnetic permeability—“the gap between the teeth”. Note that permanent magnets can be placed on the ring instead of the teeth. The ME sensor generates electrical impulses during the action of an alternating magnetic field.

The signal frequency at the ME sensor output is proportional to the speed of the crankshaft rotation. In addition, the signal sequence has a marker of the original point of reference and a quite accurate angular position of the crankshaft counted by the number of sensor electrical pulses.

## 6. Measurement Stand

The measurements were carried out using the stand shown in Figure 6. The measuring stand is the installation of a car engine with the ability to connect various nodes; for example, a crankshaft position sensor (Figure 6a).

Test bench SKAD-1 [21] consists of three components: control, engine, and measuring equipment. The control module is based on a controller Matrix MI0245 (Matrix Technology Solutions Ltd. Halifax, UK). The controller programming was performed via Flowcode7 software (Matrix Technology Solutions Ltd.). This means that several control programs can be generated according to the research of the sensors’ capabilities. An automotive engine layout is convenient for mounting and investigating the sensors, i.e., the comparative researching of the real and prototype sensors is compatible. The measuring equipment consists of a digital oscilloscope PeakTech 1195 (PeakTech Prüf- und Messtechnik GmbH, Ahrensburg, Germany) as well as optional use of a gas analyzer Kane AUTOplus 4–2 (Kane International Ltd. Hertfordshire, UK). The optional measurement of the exhaust gases is a future development for more accurate researching.

The prototypes of the ME CKP sensors were mounted to the test bench and investigated according to the preliminary generated program. The ME CKP sensor was placed near the Renault 7700103069 CKP sensor, as shown in Figure 6b. The engine was started using the Renault sensor, and the results obtained from the ME sensor were recorded and evaluated after the analysis. The purpose of this measurement was to find out the shape of the output signal, compare them with the standard shape from the Renault sensor, and check the stability of the ME sensor work. The analyzing was connected to facilitate a comparison between the output sensor signals and estimating their characteristics. The speed of rotation of the shaft pulley in the experiment was regulated from 0 up to about 3000 rpm. The sensor showed sustainable function in all operating modes. The ME sensors’ oscillograms in the low engine-speed mode and engine start mode are shown below to demonstrate the features of the output signals.

In Figure 7, two diagrams are presented. The first diagram on the left shows the sequence of the excited signals from the ME sensor located on the crankshaft pulley. The second diagram on the right clearly shows the structure of a single signal. It is significant that the signal has an oscillatory shape and requires further processing for its correct use. The left picture shows the signal of a ceramic-based sensor, and the right shows the sensor based on a lithium niobate monocrystal. The lithium niobate sensor showed significantly better results. The output voltage of the sensor was about 4 V in the case of ceramics and about 66 V in the case of lithium niobate, with the pulses fully matching the position of the pulley teeth. The test trials have shown the sustainability of the sensor working and design to mechanical vibrations, which were significant in the running engine despite the presence of the piezoelectric phase in the ME material structure. However, long-term tests in all operating modes are required to confirm the reliability of the ME CKP sensor.

## 7. Comparison of Sensor Characteristics

Comparison of the main characteristics of the standard CKP sensor used on the Renault Logan car and ME sensor will be performed below (Table 2). The Renault 7700103069 CKP sensor data were measured on the device Programmable LCR bridge HM8118. The ME CKP sensor data are based on the obtained experimental data.

Note that a direct comparison of the sensors’ characteristics cannot be considered unambiguous, since in the sensor the principle of operation uses different physical effects, and a differently toothed ring is used in the measured design. However, for an estimated comparison, such data will certainly be useful.

## 8. Conclusions

The paper considers the possibility of using ME materials for the production of CKP automobiles. An ME composite, consisting of a PZT or LiNbO_3_ piezoelectric with a size of 20 mm × 5 mm × 0.5 mm, and plates of the magnetostrictive material Metglas of the appropriate size were used as a sensitive element. The use of a composite structure based on lithium niobate made it possible to significantly increase the magnitude of the ME effect compared to the PZT/Metglas structure, increase the sensitivity of the device, reduce the noise level, and there is no need for magnetization by an external constant magnetic field. Sensors on asymmetric structures have a higher ME coefficient on the bending vibration mode; the obtained ME coefficient reached 920 V/(cm·Oe). This provides significant advantages for the application of an ME CKP sensor in automotive systems. The structure of the sensor and its connection to the CAN bus of the car were considered. Field tests at the stand showed good performance. The output voltage of the sensor was about 4 V in the case of ceramics and about 66 V in the case of lithium niobate, with the pulses fully matching the position of the pulley teeth. The studies showed good results and a high prospect for the use of ME sensors as position sensors and, in particular, of the vehicle’s CKP.

The authors are grateful for the sample of bidomain lithium niobate monocrystal with a Y + 128° provided by Andrei V. Turutin, National University of Science and Technology MISIS, RF.

## Figures and Tables

**Figure 1 sensors-20-05494-f001:**
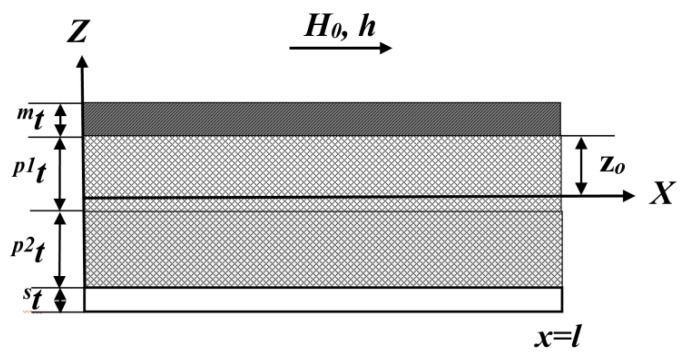
The asymmetrical ME structures.

**Figure 2 sensors-20-05494-f002:**
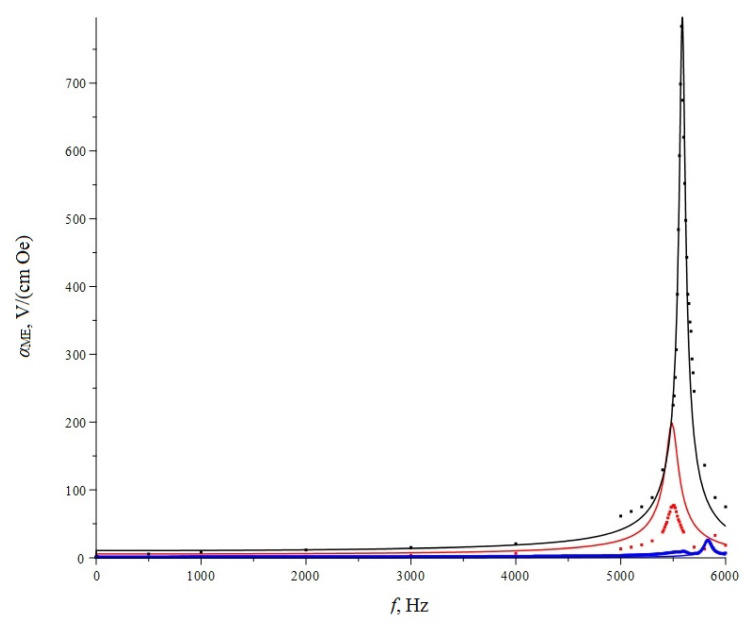
ME coefficient of the multiferroic layered structure based on LiNbO_3_/Metglas. The solid lines are theoretical curves, and the dots are experimental data (blue—symmetrical; red—asymmetrical with Cu; and black—asymmetrical with Al).

**Figure 3 sensors-20-05494-f003:**
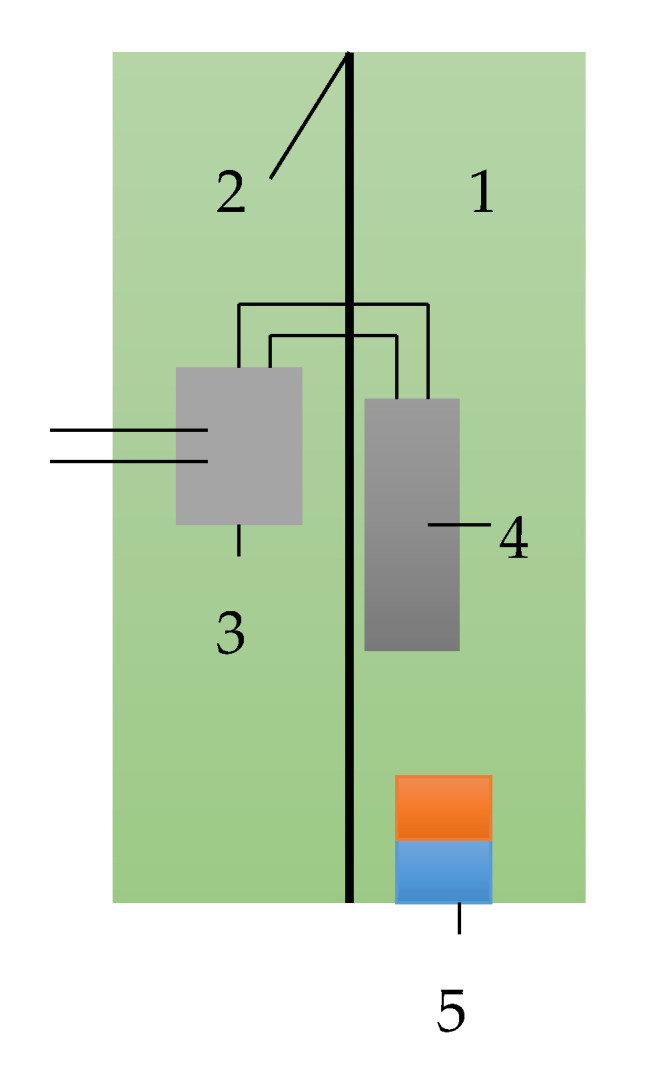
Sensor design: **1** is the sensor casing, **2** is the supporting bar, **3** is the PCB of the comparator, **4** is the ME element, and **5** is the permanent magnet.

**Figure 4 sensors-20-05494-f004:**
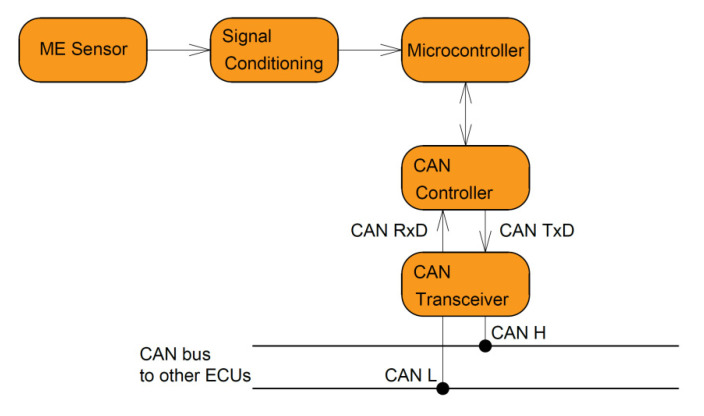
Schematic diagram of the signal processing.

**Figure 5 sensors-20-05494-f005:**
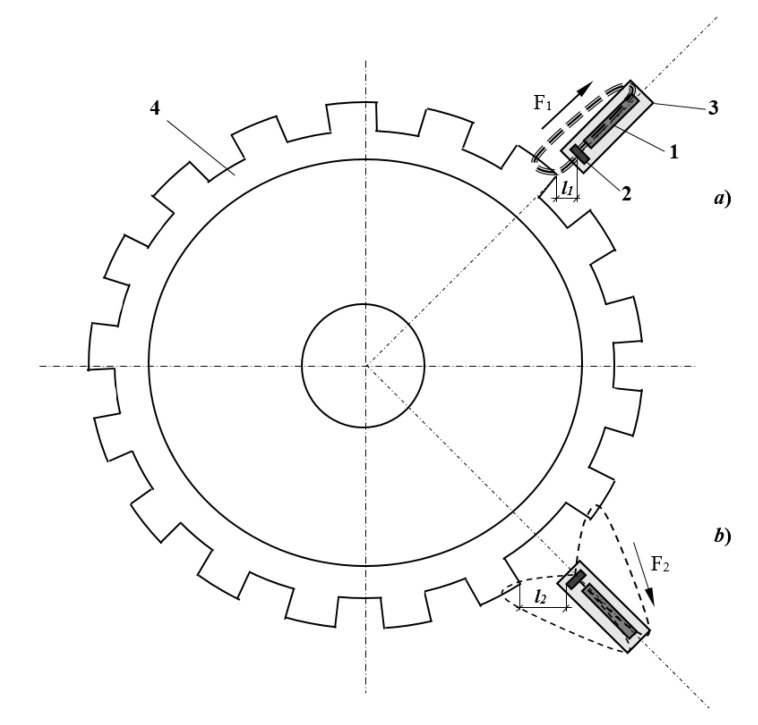
Principle of the operation of the ME CKP. (**a**) ME sensor is located above the tooth; (**b**) ME sensor is located between the teeth in the gap.

**Figure 6 sensors-20-05494-f006:**
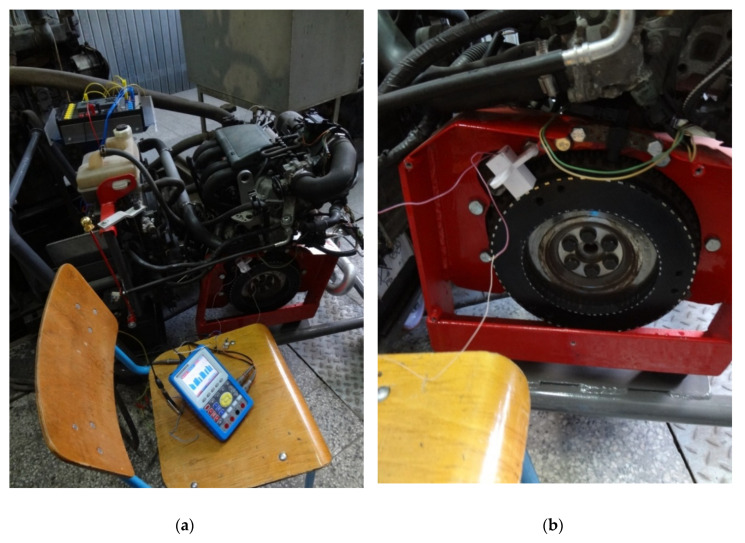
The measurement stand. (**a**) general view of the stand; (**b**) ME CKP and the crankshaft pulley.

**Figure 7 sensors-20-05494-f007:**
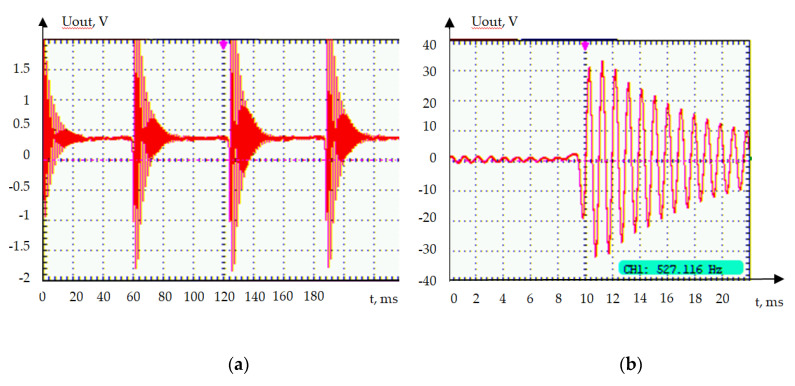
Form of the signals of the CKP sensors. (**a**) Metglas/PZT/Metglas; (**b**) Metglas/LiNbO_3_/Al.

**Table 1 sensors-20-05494-t001:** Application of magnetoelectric effect (ME) materials.

ME Devices	
Low-frequency	Magnetic field sensors, current sensors, gyrators, energy harvester, position sensors, etc.
Microwave	Phase shifters, resonators, attenuators, filters, etc.

**Table 2 sensors-20-05494-t002:** Approximate technical characteristics of the CKP sensors.

	Renault 7700103069 CKP Sensor	ME CKP Sensor
1. The range of winding resistance	403 Ohm	-
2. The sensor resistance	-	>1 MOhm
3. The inductance of the winding	94 mH	-
4. ME coefficient on the bending mode	-	920 V/(cm·Oe)
5. Minimum sensor voltage amplitude	0.28 V	1.5 V
6. Maximum sensor voltage amplitude	222 V	66 V
7. The overall dimensions	25 mm × 55 mm × 90 mm	25 mm × 45 mm × 90 mm
8. The weight	10 g	8 g

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
