# Peer review of "A Magnetoelectric Automotive Crankshaft Position Sensor"

_sensors, 2020, doi:10.3390/s20195494_

Round 1

Reviewer 1 Report

This article discusses the new type of magnetoelectric automotive crankshaft position sensor. The topic considered in the article seems to be very interesting from the practical point of view. In particular, the authors propose a sensor based on the magnetoelectric effect, which can be applied in the automotive industry. A great advantage of the work is a measurement stand developed by the authors based on a car engine with the ability to connect various nodes.

However, this article has a number of significant disadvantages.

  1. The principle used for measuring position by magnetic field pulses is well known. At present, such sensors are mostly based on the Hall effect, and on giant magnetoresistance. However, the authors did not provide any references to other authors.

https://www.youtube.com/watch?v=Exz4f4RfBFY

  1. Magnetoelectric effect in PZT/Metglas and LiNb03/Metglas structures is well known. For example see following articles:

- Magnetoelectric metglas/bidomain y + 140º-cut lithium niobate composite for sensing fT magnetic fields APPLIED PHYSICS LETTERS 112, 262906 (2018)

 - Magnetoelectric Effect in the Bidomain Lithium Niobate/Nickel/Metglas Gradient Structure Phys. Status Solidi B 2019, 1900398

No references are also given. 

  1. The article repeats author’s publications of previous years. Most of the figures are taken from other papers. In particular, figures 1, 3, 4. In addition, the figures themselves and their captions should be modified. Not all elements from figures are defined. See figures 1 and 6.
  2. The article is poorly structured. It is difficult to understand what magnetoelectric structures were investigated, since the authors do not provide their detailed description and manufacturing technology in the article. The need for a theoretical part in this article is unclear. In addition, the experimental results are not discussed.
  3. Given theory has been previously described in other articles. Moreover, it is impossible to use the formulas given in the article for calculations, because some of the variables are not defined in the article.
  4. The authors indicate that the dimensions of the sensor were chosen similar to those of the VC-CS 0112 sensor. At the same time, a comparison of characteristics was made with the Renault Logan CKP sensor. Authors do not explain the reason for that. In addition, authors do not give adequate description for that sensor.
  5. The references provided in this article do not describe well the current state in the field. References to articles of other authors are not given. For example:

-  Design and Development of a Tachometer Using Magnetoelectric Composite as Magnetic Field Sensor IEEE TRANSACTIONS ON MAGNETICS  DOI 10.1109/TMAG.2017.2721900

- An angle sensor based on magnetoelectric effect Sensors and Actuators A 262 (2017) 108–113

- A new angular velocity sensor with ultrahigh resolution using magnetoelectric effect under the principle of Coriolis force Sensors and Actuators A 238 (2016) 234–239

- Magnetoelectric laminate composite based tachometer for harsh environment applications Applied Physics Letters 91, 122904 (2007); doi: 10.1063/1.2784959.

Authors should remove references to sources without translation into English. See ref. 6., ref. 7 etc.

  1. Authors define structures as asymmetrical when magnetic layers are located on one side of the piezoelectric. While in the literature, structures are called asymmetrical when magnetostriction coefficients of the magnetic layers on both sides of the piezoelectric layers differ in sign. For example, structure Metglas/PZT/Ni.
  2. Authors write that the proposed sensor is repairable, which is an advantage, as well as low cost. But they do not give any proof of this. It is also stated that the applied solutions allow achieving low noise levels. It would be useful if authors could give some noise measurements.
  3. The article also does not cover issues such as the operating temperature range of the sensor or the operating speed range.
  4. Also there is a feeling that english should be improved.

In general, despite the interesting research topic, this article requires a lot of revision. Both structure and results description should be sufficiently improved. Authors should be more accurate in describing structures and experimental results.

Author Response

Dear colleague, thank you very much for your comments. We tried to improve the article. Here are the answers to your comments.

  1. We have included links to articles [6-9].
  2. We've included the link [19].
  3. We have changed the figures.
  4. We have structured the article, and also added the discussion of experimental results (lines 155-172).
  5. The theory of the magnetoelectric effect in the field of electromechanical resonance in the bending mode depends significantly on the boundary conditions determined by the fixation of the magnetoelectric sample. In this article, we studied a rather non-trivial fastening: one end is tightly clamped, and the other is freely supported. A three-layer structure containing a lower passive layer is also studied in one of the variants. Therefore, the final formula for the ME voltage coefficient has not been obtained in any of the previous articles. Certainly, in order for this formula to be used for calculations, we had to provide all the necessary concomitant formulas in the article. Since, in General, quite a lot of articles on the magnetoelectric effect in the field of Electromechanical resonance on a bending mode have already appeared, some of the concomitant formulas given above repeat those previously given in other articles.

The text of paragraph 2 was improved.

  1. Different sensors were measured during the research. We specified the specific type in the article (lines 180, 191, 287, 290).
  2. We have revised the links.
  3. We have explained the definition of an asymmetric structure for this article (line 90-93).
  4. Disputed statements were excluded from the text of the article. Information about the measurement noise are given in the line 169-170.
  5. The range of temperatures seen in the line 161-172. The speed range is discussed in lines 168-169.
  6. We tried to improve English in the article.

Graying text highlighting is a change in the structure, i.e., transferring text from another paragraph. The yellow text highlighting is a change or addition of text.

Reviewer 2 Report

Please find my comments in the attached file.

Author Response

Dear colleague, thank you very much for your comments. We tried to improve the article. Here are the answers to your comments.

Comment:Interval between two pulses is about 60 ms which would correspond to 16 rpm. I have a big doubt that any petrol/gasoline engine can be run in a stable way at 16 rpm. This makes me thinking that the experiment wasn’t done on the running engine but was just emulated by applying external force. Moreover, intrinsic decay of the oscillations from the sensor is about 30 ms which is too inertial to be used in automotive applications. Even more, the excitation frequency is shown here corresponds to 500 Hz or 2ms of periodicity which is again too slow.

Reply: The measurement mode for figure 7 is discussed in lines 265-271.

Comment:Timing performance and comparison to usual inductive sensor.

Reply: We conducted a comparison with an inductive sensor in lines 192-193.

Comment:Temperature stability of the sensor and comparison to a usual inductive sensor.

Reply: Temperature characteristics are discussed in lines 161-172.

Comment:Stability against mechanical vibration which significant in the running engine. You use piezoelectric. What are the consequences???

Reply: Stability against mechanical vibration is discussed in lines 181-184.

Comment:General robustness of design: inductive sensor has coil, ferromagnetic core and permanent magnet. Usual failure there is mechanical break of the coil wire. You propose to have on board in your sensor: coil, multilayered sensor, electronic circuit, permanent magnet. How can you ensure that it will be a reliable solution for many years of work?

Reply: General robustness of design is discussed in lines 192-200.

Graying text highlighting is a change in the structure, i.e., transferring text from another paragraph. The yellow text highlighting is a change or addition of text.

Reviewer 3 Report

The paper describes a new position sensor for automotive applications. It is interesting because it does not require a magnetic field for biasing and because of the high signal levels obtained.

The paper would be more useful if the section with the principles of operation were placed close to the beginning of the paper. 

There must be more discussion of the comparison between the fabricated sensor and the ceramic sensor.

All of the references are to the group's own papers. It is absolutely necessary to find other literature dealing with the subject at hand and to provide the appropriate references. The group's own papers cannot be more than ~20% of the total number of references.

Author Response

Dear colleague, thank you very much for your comments. We tried to improve the article. Here are the answers to your comments.

  1. We placed a section with the principles of operation were placed close to the beginning of the paper.
  2. The comparison between the monocrystal sensor and the ceramic sensor is shown in lines 155-172.
  3. We have revised references.

Graying text highlighting is a change in the structure, i.e., transferring text from another paragraph. The yellow text highlighting is a change or addition of text.

Reviewer 4 Report

Review of "Magnetoelectric automotive crankshaft position sensor"
for Sensors

The authors have put together a thorough study of the possibility for applying magnetoelectric magnetization sensors to the application of an internal combustion engine flywheel crankshaft position sensor (CPS). The authors consider the composite of PZT or LiNbO3 for the piezoelectric combined with Metglas for the magnetostrictive element. While perceived to be a small improvement, the benefits of a magnetoelectric sensor over hall effect or magnetization induction sensors are discussed here. The technical details of the article are sound and the potential impact to the community is high, however i feel the following improvements need to be made before the article could be ready for publication.

CPS sensors are notorious for degradation over the life of the vehicle. Does an ME sensor address the degradation effects of traditional parts?

The authors should discuss more the benefit of lower power consumption in these applications. While weight savings and increased sensitivity are beneficial, the rise of hybrid power systems and increased efficiency demands require all sensors and systems in automotive to gain in efficiency. ME sensors do not need electric current to operate, and piezo impedance-read voltages are very low. This should be discussed, and i feel highlighted as a stronger advantage.

Figure 7 needs to be properly plotted. These seem to be images from an oscilloscope. However a plot with x-y axis labels should be created that are clear for the reader. Extra information from the oscilloscope display is also a distraction.

I suggest the following recent papers for the added applications of magnetoelectric sensors in MEMs technology:

N. Lukat, R.-M. Friedrich, B. Spetzler, C. Kirchhof, C. Arndt, L. Thormälen, F. Faupel, C. Selhuber-Unkel, Mapping of magnetic nanoparticles and cells using thin film magnetoelectric sensors based on the delta-E effect Sens. Actuators A, 309. 112023, (2020), https://doi.org/10.1016/j.sna.2020.112023

Magnetic field response of doubly clamped magnetoelectric microelectromechanical AlN-FeCo resonators
SP Bennett, JW Baldwin, M Staruch, BR Matis, J LaComb, OMJ van't Erve, ...
Applied Physics Letters 111 (25), 252903

Author Response

Dear colleague, thank you very much for your comments. We tried to improve the article. Here are the answers to your comments.

  1. The effects of degradation are discussed in the lines 192-194.
  2. The advantage of ME sensor energy consumption are considered in lines 194-196.
  3. We corrected the figure 7.
  4. We added references.

Graying text highlighting is a change in the structure, i.e., transferring text from another paragraph. The yellow text highlighting is a change or addition of text.

Round 2

Reviewer 1 Report

The content and format of the article was improved.
Some of the drawbacks were fixed. 

Author Response

Dear colleague,

Thank you very much for your help in improving the paper. The authors team is grateful for your comments and suggestions, which allowed us to better describe our research.

Reviewer 2 Report

The revised manuscript differs from the initial one only by small corrections and do not address the main issues I have enlisted in my previous report. Therefore I still susggest the Editor to reject it without further resubmission.

As I described in my first report, from the time interval between two
pulses one can easily conclude that the experimental data the authors show don't correspond anyhow to a really working engine as they claim.
Moreover, the temporal response diagram suggests that the sensor is very inertial. Again, simple estimations which I have done in my report show that this sensor is not anyhow suitable for the purpose of a crankshaft sensor.

Author Response

Dear colleague,

Thank you very much for your help in improving the paper. The authors team is grateful for your comments and suggestions, which allowed us to better describe our research.

We have explained the procedure for conducting the experiment that caused you doubts (lines 268-272). The resulting waveform can be converted to an acceptable one electronically.

The yellow text highlighting is a change or addition of text.

Reviewer 3 Report

The authors have added a significant number of references to the work of other groups. They could have added more modern references.

The English requires some attention.

I still think the paper would flow more logically if the Principle of Operation were placed near the beginning of the paper, perhaps right after the Introduction, than when it is placed close to the end of the paper.

Author Response

Dear colleague,

Thank you very much for your help in improving the paper. The authors team is grateful for your comments and suggestions, which allowed us to better describe our research.

We will improve English in our paper. At the same time, the authors believe that the paper structure can be left unchanged.